# Is night-time light intensity associated with cardiovascular disease risk factors among adults in early-stage urbanisation in South India? A cross-sectional study of the Andhra Pradesh Children and Parents Study

Tina Bonde Sorensen [1], Robin Wilson,[2] John Gregson,[3] Bhavani Shankar,[4] Alan D Dangour,[1] Sanjay Kinra[5]

► Prepublication history and additional materials for this paper is available online. To view these files, please visit the journal online (http://dx.doi.org/10.1136/bmjopen-2019-036213).

For numbered affiliations see end of article.

**Correspondence to**
Tina Bonde Sorensen;
tinabondesoerensen@gmail.com

## ABSTRACT

**Objectives** To explore associations of night-time light intensity (NTLI), a novel proxy for continuous urbanisation levels, with mean systolic blood pressure (SBP), body mass index (BMI), fasting serum low-density lipoprotein (LDL) and fasting plasma glucose (FPG), among adults in early-stage urbanisation in Telangana, South India.

**Design** Cross-sectional analysis of the third wave of the Andhra Pradesh Children and Parents Study cohort.

**Setting** 28 villages representing a continuum of urbanisation levels, ranging from rural settlement to medium-sized town in Telangana, South India.

**Participants** Data were available from 6944 participants, 6236 of whom were eligible after excluding pregnant women, participants younger than 18 years of age and participants missing data for age. Participants were excluded if they did not provide fasting blood samples, had implausible or missing outcome values, were medicated for hypertension or diabetes or had triglyceride levels invalidating derived LDL. The analysis included 5924 participants for BMI, 5752 participants for SBP, 5287 participants for LDL and 5328 participants for FPG.

**Results** Increasing NTLI was positively associated with mean BMI, SBP and LDL but not FPG. Adjusted mean differences across the range of village-level NTLI were 1.0 kg/m$^2$ (95% CI 0.01 to 1.9) for BMI; 4.2 mm Hg (95% CI 1.0 to 7.4) for SBP; 0.3 mmol/L (95% CI −0.01 to 0.7) for LDL; and −0.01 mmol/L (95% CI −0.4 to 0.4) for FPG. Associations of NTLI with BMI and SBP were stronger in older age groups.

**Conclusion** The association of NTLI with cardiovascular disease (CVD) risk factors identify NTLI as a potentially important tool for exploring urbanisation-related health. Consistent associations of moderate increases in urbanisation levels with important CVD risk factors warrant prevention strategies to curb expected large public health impacts from continued and rapid urbanisation in India.

## BACKGROUND

Cardiovascular diseases (CVDs), primarily coronary heart disease, are the leading causes

### Strengths and limitations of this study

► We used a novel, standardised proxy of continuous urbanisation levels, remote sensing data on night-time light intensity, measured at the village level.
► The villages had experienced rapid uneven urbanisation during the decades preceding the study and represented a continuum of urbanisation levels ranging from a rural settlement to a medium-sized town.
► The study used a large population-based sample from 28 urbanising villages at early stages of urbanisation.
► Differences between excluded and included participants warranted some caution of generalising the findings to the general population.
► The cross-sectional design prevented us from making causal inferences or exploring the impact of urbanisation rate on cardiovascular diseases risk factors.

of death (28%) and disability-adjusted life years (14%) in India.[1] They occur 5–10 years earlier than in Western populations,[2] and the age-standardised mortality rate (272 per 100 000 persons) exceeds the global average and rates of some high-income countries.[3 4] Individuals and societies in India therefore continue to experience substantial losses of productivity and income.[1 5–7]

### Urbanisation and CVD risk factors

Urbanisation is often described as a key driver of CVDs in low-income and middle-income (LMICs) acting on CVDs and their risk factors through inter-related changes in social, physical and build environments, sociodemographics, lifestyle and mental health. Some of these change are associated with harmful

impacts on cardiovascular health, for example, due to reduced physical activity and nutrition transitions, while others may be beneficial, for example, through more and better education and access to health and social services.[8–12] A number of simple rural–urban comparisons and migration studies provide insight to the association of urbanisation level with CVD risk factors, such as overweight and obesity, hypertension, diabetes and dyslipidaemia.[8 13–21] The broad rural–urban dichotomies are useful, however insufficient, for exploring which factors of urbanisation are most important for health in LMICs.[22] This is in part because no universal definition exists of 'rural' and 'urban' environments, and data on neighbourhood characteristics are often not available from these settings.[22] Furthermore, residents' characteristics tend to be more similar within than between urban and rural areas. For example, urban residents may on average enjoy wider access to goods and services, higher levels of education, occupation, affluence and so on, whereas the opposite may be the case in rural areas.[22 23] Such limited discordance at either end of the urbanisation continuum makes it difficult to tease out the underlying mechanisms of observed differences in CVD risk between the two extremes. Urbanisation factors may change at different rates with urbanisation,[22] however, and thus vary widely across the continuum from rural to urban areas. It may therefore be more useful and informative to study the impact of urbanisation level on CVD risk factors in transitioning populations, that is, populations representing various levels of urbanisation across the continuum, where the underlying mechanisms may be easier to separate. Unfortunately, it is challenging to accurately measure urbanisation as a continuum, whether using single-component (eg, population density) or multicomponent classification systems (eg, housing types and densities, economic activities, physical environment and services).[22] This is particularly true in LMICs where these types of data are often scarce.

### Night-time light intensity (NTLI) as a measure of urbanisation level

As an alternative to physical data-based scales, remote sensing data have been widely used to characterise urban landscapes and scale of urbanisation.[24 25] The NTLI data, obtained by the USA's Defence Meteorological Satellite Programme's Operational Linescan System, are suggested a valid proxy for urbanisation level and dynamics due to their strong and consistent relationships with demographic and economic indicators of urbanisation such as urban build up area, population density, economic activity and energy consumption at global, regional and local levels.[24–27] The NTLI data, which have near global coverage and a freely available historical data archive dating back to 1992,[28] offer a unique opportunity to study the association of urbanisation, measured by a single standard measure worldwide, with CVD risk.

### Study aim to address evidence gaps

This study aimed to explore associations of urbanisation level (measured by a continuum of NTLI) with mean levels of four leading risk factors for CVDs, systolic blood pressure (SBP), body mass index (BMI), fasting serum low-density lipoprotein (LDL) and fasting plasma glucose (FPG), in adults from 28 villages in rapidly urbanising Telangana, South India. Improved understanding of the association of urbanisation level and CVD risk factors in settings that are transitioning from rural areas to greater levels of urbanisation will help inform policy for disease prevention and control. Such policies are imperative in India, where economic growth and urbanisation continues, to avoid further adverse economic and social impacts of (premature) CVD morbidity and mortality.

## MATERIALS AND METHODS
### Study design, setting and participants

The current cross-sectional study analysed data from the third and most recent survey wave of the Andhra Pradesh Children and Parents Study (APCAPS)[29] and was set in 28 villages near Hyderabad in Telangana, South India. The APCAPS birth cohort was originally established (in 2003–2005) to investigate the short-term and long-term effects of early life nutrition on CVDs and their risk factors as a long-term follow-up of the Hyderabad Nutrition Trial (HNT). The HNT (1987–1990) was a cluster-randomised community trial, in which all pregnant women and young children received (15 villages) or awaited (14 villages) a nutrition intervention. The APCAPS villages experienced uneven rates of urbanisation over time, and from the second survey wave, the cohort was expanded to include more participants and information on CVD risk factors to investigate diverse environmental, lifestyle and metabolic risk factors for CVDs in the birth cohort as well as among the children's parents and siblings.[30 31] The third and most recent data wave (conducted in 2010–2012, n=6944 [$n_{(index children)}$=1360; $n_{(mothers)}$=1379; $n_{(fathers)}$=1140; $n_{(siblings)}$=2344; $n_{(general population)}$=721]) was selected for cross-sectional analysis due to its large adult population (≥18 years), wide age range and varying levels of urbanisation. Details on the recruitment of APCAPS participants from the first to the third wave of the APCAPS are presented in figure 1. Further details on the cohort are reported elsewere.[30 31]

### Sample size and selection

All non-pregnant adults (18 years or older) from the third APCAPS survey wave were eligible for inclusion (n=6236/6944) (figure 1). Participants who reported taking medication for outcome-related conditions were excluded to reduce the risk of biasing results, for example, towards the null if participants with medically controlled blood pressure sustained high levels of risk factors for hypertension. Participants were excluded from analyses of glucose if they reported fasting for less than 8 hours at the time of giving the blood (n=366) or were currently

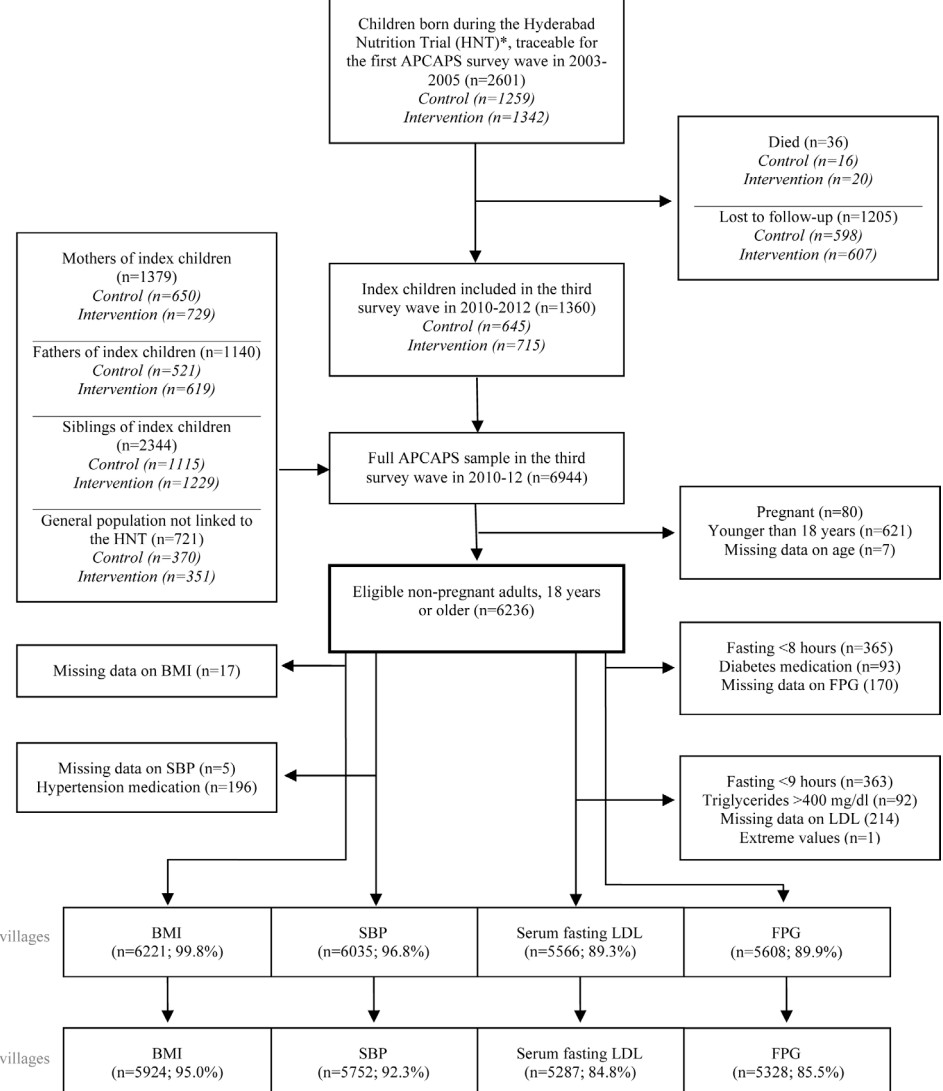

**Figure 1** Sampling and participant flow for the third survey wave of the APCAPS. *The Hyderabad Nutrition Trial (1987–1990) included 15 intervention and 14 control villages from the Integrated Child Development Services Scheme (stepped-wedge cluster randomised nutrition supplementation trial). APCAPS, Andhra Pradesh Children and Parents Study; BMI, body mass index; FPG, fasting plasma glucose; LDL, low-density lipoprotein; SBP, systolic blood pressure.

taking medication (tablets or insulin) for diabetes (n=94). Participants were excluded from analysis of LDL if they reported fasting for less than 9 hours (n=364) or had triglyceride levels above 400 mg/dL[32] (n=92). Participants who reported taking antihypertensive medication were excluded from analyses of SBP (n=197) (figure 1). One of the 28 surveyed villages was significantly larger and had a considerably higher NTLI value than the other villages, and its participants were excluded (n=297) to avoid over-reliance of results from this outlier village. The final sample sizes by study outcome are presented in figure 1.

## Data
### Data collection and procedures
Trained field staff collected data on health and sociodemographic information in the local language, Telugu, using semistructured questionnaires. Trained field staff

and medical doctors performed the clinical assessments in local clinics and at the National Institute of Nutrition in Hyderabad. Venous fasting (8–12 hours) blood samples were collected, separated within 30 min and stored locally at −20°C. Biochemical assays were performed using the Cobas311 autoanalyser at the Genetics and Biochemistry Laboratory at The South Asia Network for Chronic Diseases of the Public Health Foundation of India, New Delhi. To define APCAPS village boundaries, trained field staff performed aerial tracing of Bing satellite imagery, using Open Street Map software and GPS-based ground surveying during 2012–2013.

The NTLI data, which is collected by the USA's Defence Meteorological Satellite Programme's Operational Linescan System, was obtained online from the National Oceanic and Atmospheric administration,[33] through the Google Earth Engine.[34] The raw satellite data is processed

into annual averages of the intensity of light emitted from the Earth's surface at night from persistent sources associated with human settlement, that is, excluding transient light sources such as lightning, fire, aurora, solar and lunar light. The processed data are aggregated into 30 arc seconds grids (equivalent to a resolution of approximately $1\,km^2$ pixels)[35] and published as unitless digital numbers ranging from 0 (no light) to 63 (light saturation).[24 27] We obtained the NTLI imagery.

### Defining urbanisation level

Urbanisation is commonly described as the process and dynamics of permanent concentration of a population in urban settlements and/or the adaption of lifestyles associated with urban environments, for example, sedentary lifestyle or nutrition trantision.[10 11 36 37] Although internal migration to urban areas is common in India, most urbanisation is reported to be driven by the natural growth and development of rural areas and their subsequent reclassification into urban settlements.[38] The term 'early-stage urbanisation' represents the early transition of rural settlements towards more urbanised physical environments and ways of life in the absence of development into large towns or cities. For the purpose of this study, the term 'urbanisation level' defines the level of urbanisation of a village at the time of survey in 2010–2012, as measured by NTLI (described in the following section).

### Determining urbanisation levels of APCAPS villages

The data defining village boundaries were overlaid with the NTLI grids from 2011 (technical issues with satellite sensors prevented the use of NTLI data from 2012).[28] Twenty-four villages were intersected by two or more pixels, which in some cases had different NTLI values. Therefore, to obtain one NTLI value (urbanisation level) per village, we used a super-resolution method to resample the NTLI images to a higher resolution, before summing the unitless digital numbers of pixels within or intersecting with the village boundary to produce an overall village level. This ensured giving the appropriate weighting to each of the pixel values present over each village. Finally, the data were calibrated using the ridgeline sampling regression methods,[28] and oversaturation of light was removed manually. Since the first survey wave in 2003–2005, the 28 APCAPS villages had changed from rural agricultural communities with similar NTLI levels to represent a continuum of NTLI levels with growing differentiation at higher levels.

### Selection of study outcomes

For this analysis, we prespecified the four outcomes, BMI, SBP, LDL and FPG, from a number of CVD risk factors available in the APCAPS, due to their well-established and strong associations with CVDs and their modifiable nature.[32 39] The selection of continuous outcomes also enabled exploration of linear relationships between increasing level of NTLI and the CVD risk factors.

### Measurement of study outcomes

APCAPS cohort data were collected using standardised approaches. Weight was measured twice to the nearest 0.1 kg with digital weight scales (Seca Leicester 899; Chasmors Ltd, London, UK) while wearing light clothing and no shoes. Standing height was measured twice to the nearest 0.1 cm with a portable stadiometer (Seca Leicester Height Measure; Chasmors Ltd, London, UK). The two weight and height measurements were averaged, and BMI was calculated for the analyses as weight (kg)/ height $(m)^2$. Blood pressure was measured three times at the right arm in the sitting position using a digital device and appropriate sized cuff (Omron hem 7300; Matsusaka Co, Japan). The last two measurements of SBP were averaged for the analysis. FPG was assessed on the day of sampling by enzymatic glucose oxidase/peroxidase-4-aminophenazone-phenol method (Randox Laboratories; London, UK). Fasting LDL was derived indirectly using the Fridewald-Fredrickson formula: LDL = total cholesterol − high-density lipoprotein − triglycerides/5 (all measures in mg/dL).[32 40]

### Biochemical assays of variables needed to derive study outcomes

Serum total cholesterol and triglycerides were assessed by enzymatic methods using cholesterol oxidase-peroxidase-4-aminophenazone-phenol and glycerol phosphate oxidase-peroxidase-4-aminophenazone-phenol (Roche Diagnostics GmbH; Mannheim, Germany). High-density lipoprotein (used for derivation of LDL) was derived by direct method using polyethylene glycol modified cholesterol esterase, cholesterol oxidase and dextran sulphate (Roche Diagnostics GmbH; Mannheim, Germany).

### Quality assurance

Trained field staff and medical doctors followed standardised procedures, and anthropometric equipment were calibrated at the start of every clinic. Reproducibility of clinical measurements were evaluated by repeating measures on a 5% random subsample after 1–3 weeks. The consistency was high for all measures with intraclass correlation coefficient for anthropometric measurements >0.98, blood pressure >0.85 and biochemical assays >0.940.[41] The quality of biochemical assays were monitored by the Cardiac Biochemistry laboratory at the All India Institute of Medical Sciences, which participates in the UK National External Quality Assessment Programme and the External Quality Assessment Scheme of Randox International.

### Patient and public involvement

Study participants (patients) and the public were not involved in the design or planning of the study.

### Statistical methods

Variables were assessed for outliers, cleaned accordingly and examined for normality. NTLI data were nonnormally distributed and log-transformed for analysis. To preserve study power, we allowed sample sizes to vary by outcomes. Mixed effects linear regression models with

clustering by household and village were fitted in all analyses to account for the hierarchical nature of the data and the sampling strategy. NTLI (log transformed), BMI, SBP, LDL, FPG, age and room temperature at clinical assessment (the latter for analysis of SBP) were included as continuous variables. Gender (women or men), caste (general caste, scheduled caste, scheduled tribe, other backward class and other), religion (Muslim, Hindu, Christian and other), current marital status (married or unmarried), season of survey and season of birth (summer (March–May), South West monsoon (June–October), winter (November–February)) were included as categorical variables. Study characteristics were presented by thirds of ranked NTLI, referred to as low, medium and high levels. We evaluated potential confounders for each outcome using the double-selection method.[42] One final set of covariates, identified as potential confounders for at least one exposure–outcome relationship, were used in analyses of all four outcomes. Level of urbanisation may affect CVD risk profiles of different genders and age groups differently due to, for example, gender inequality, employment patterns and readiness to adopt new lifestyles and behaviours.[23 43–45] We therefore tested for interaction by gender and age. We did not consider sociodemographic factors such as education, occupation and socioeconomic status as potential confounders or effect modifiers due to their importance on the causal pathway from urbanisation level to CVD risk factors. We did not adjust the main analysis of SBP for room temperature at clinical assessment because a large proportion of data was missing for this variable ($n_{(missing)}$=2648 (45%)). However, a supplementary analysis for room temperature-adjusted SBP was presented in the supplementary material.

A sensitivity analysis was performed to assess the influence of trial status among index children from the HNT. A second sensitivity analysis was performed to evaluate whether borderline outlier villages (with higher or lower observed mean outcomes than the majority of villages) influenced regression estimates considerably. Differences between included and excluded participants were examined with $\chi^2$ and linear and logistic mixed effects models with clustering by village and household. All analyses were performed in Stata V.14.

## RESULTS

### Study characteristics

A total of 6236 non-pregnant adults were eligible for inclusion in the current study (90% of the third APCAPS survey wave). After removing extremes and participants with missing outcome data for specific analyses, the final samples came to 5924 participants for BMI, 5752 participants for SBP, 5287 participants for LDL and 5328 participants for FPG (figure 1). Approximately half of participants were women (table 1). The overall median age was 32 years (IQR 24–48), with women on average older than men. Men and women most frequently identified with other backward caste and Hindu religion.

Women were more likely to be married (76%) than men (59%). Village-level NTLI ranged from 61.7 (equivalent to 4.1 on the log scale) to 1081.1 (equivalent to 7.0 on the log scale). The distribution of sociodemographic characteristics were similar across low, medium and high level of NTLI. None of the included (non-medicated) participants reported ever being diagnosed with heart disease, including stroke.

### Crude and adjusted association of NTLI with CVD risk factors

Model predicted mean BMI, SBP and LDL increased with rising NTLI in crude and adjusted linear models (figure 2 and table 2). Across the range of NTLI, the fully adjusted predicted mean increased $1.0\,kg/m^2$ (95% CI 0.01 to 1.9, p=0.05) for BMI and 4.2 mm Hg (95% CI 1.0 to 7.4, p=0.01) for SBP. The pattern was similar for LDL with a fully adjusted predicted mean increase across the range of NTLI values of 0.3 mmol/L (95% CI −0.01 to 0.7), although the evidence was weak from crude (p=0.06) and fully adjusted linear models (p=0.06). There were no linear association of NTLI with FPG (fully adjusted mean difference across the range of NTLI: −0.01 mmol/L (95% CI −0.4 to 0.4, p=0.97).

Model predicted mean SBP was on average 6.1 mm Hg (95% CI 5.42 to 6.9, p<0.001) higher among men than women, whereas levels of BMI, LDL and FPG did not differ by gender. However, there was no statistical evidence of gender modifying any of the associations between NTLI and the CVD risk factors. There was strong evidence that age modified the associations of NTLI with BMI (p<0.001) and SBP (p<0.001), but not LDL or FPG. NTLI was not associated with BMI or SBP in the younger age groups (up to 39 years), whereas the evidence and magnitudes of associations from both crude and fully adjusted models were stronger in older age groups (from 40 years) than predicted in the overall analysis (see online supplemental figure 1 and table 1). Online supplemental figure 2 illustrates at which levels of NTLI the fully adjusted models predicted the greatest probability of associations differing by age group. Overall and age-specific results of a supplementary analysis for SBP adjusted for room temperature at clinical assessment (n=3160) were similar to the main and age-specific SBP analyses (see online supplemental tables 2, 3 and figure 3).

### Comparing included and excluded participants

Participants who were excluded from the main analysis due to missing data, medication use, or high triglyceride levels differed in some respects from the analysed participants. Excluded individuals were on average older, less likely to attend clinical assessment/interview in the summer and had higher levels of the four CVD risk factors than included individuals.

### Sensitivity analyses

A sensitivity analysis among HNT index children (n=1245 (21% of overall sample); intervention n=622, control n=623) did not indicate confounding or interaction by

**Table 1** Participant characteristics of APCAPS adults from 27 villages, 2010–2012 (n=5932)

| | NTLI*, women (n=2753 (46%)) | | | NTLI*, men (n=3179 (54%)) | | |
|---|---|---|---|---|---|---|
| | **Low** | **Medium** | **High** | **Low** | **Medium** | **High** |
| Night-time light intensity (NTLI), median (IQR) | 138.2 (121–149) | 202.4 (185–273) | 492.8 (485–753) | 138.2 (129–149) | 202.4 (185–273) | 492.8 (396–753) |
| Age, median (IQR) | 40 (25–46) | 40 (25–46) | 38 (25–45) | 28 (23–51) | 29 (24–50) | 28 (23–49) |
| Age group (years), n (%) | | | | | | |
| <30 | 299 (38.7) | 372 (38.2) | 416 (41.3) | 489 (54.5) | 595 (52.7) | 636 (55.2) |
| 30–39 | 85 (11.0) | 106 (10.9) | 134 (13.3) | 57 (6.4) | 74 (6.6) | 91 (7.9) |
| 40–49 | 261 (33.8) | 348 (35.8) | 348 (34.5) | 98 (10.9) | 151 (13.4) | 154 (13.4) |
| 50+ | 127 (16.5) | 147 (15.1) | 110 (10.9) | 254 (28.3) | 309 (27.4) | 271 (23.5) |
| Total | 772 (100) | 973 (100) | 1008 (100) | 898 (100) | 1129 (100) | 1152 (100) |
| Caste, n (%) | | | | | | |
| General caste | 51 (6.6) | 79 (9.2) | 76 (7.5) | 51 (5.7) | 89 (8.9) | 63 (5.5) |
| Scheduled caste | 277 (35.9) | 375 (43.6) | 292 (29.0) | 331 (36.9) | 428 (42.6) | 335 (29.1) |
| Scheduled tribe | 9 (1.2) | 3 (0.4) | 5 (0.5) | 11 (1.2) | 3 (0.3) | 7 (0.6) |
| Other backward | 425 (55.1) | 374 (43.5) | 628 (62.3) | 488 (54.3) | 455 (45.3) | 739 (64.2) |
| Other | 9 (1.2) | 29 (3.4) | 7 (0.7) | 17 (1.9) | 30 (3.0) | 8 (0.7) |
| Total | 771 (100) | 860 (100) | 1008 (100) | 898 (100) | 1005 (100) | 1152 (100) |
| Religion, n (%) | | | | | | |
| Muslim | 19 (2.5) | 37 (4.3) | 71 (7.0) | 16 (1.8) | 45 (4.5) | 64 (5.6) |
| Hindu | 742 (96.2) | 786 (91.4) | 919 (91.2) | 875 (97.4) | 931 (92.6) | 1079 (93.7) |
| Christian | 9 (1.2) | 37 (4.3) | 16 (1.6) | 7 (0.8) | 29 (2.9) | 8 (0.7) |
| Other | 1 (0.1) | 0 (0) | 2 (0.2) | 0 (0) | 0 (0) | 1 (0.1) |
| Total | 771 (100) | 860 (100) | 1008 (100) | 898 (100) | 1005 (100) | 1152 (100) |
| Marital status, n (%) | | | | | | |
| Not married | 196 (25.4) | 238 (24.5) | 229 (22.7) | 381 (42.4) | 459 (40.7) | 487 (42.3) |
| Married | 576 (74.6) | 734 (75.5) | 779 (77.3) | 517 (57.6) | 669 (59.3) | 665 (57.7) |
| Total | 772 (100) | 972 (100) | 1008 (100) | 898 (100) | 1128 (100) | 1152 (100) |
| Hyderabad Nutrition Trial, n (%) | | | | | | |
| Control | 643 (83.3) | 826 (84.9) | 835 (82.8) | 667 (74.3) | 852 (75.5) | 865 (75.1) |
| Intervention | 129 (16.7) | 147 (15.1) | 173 (17.2) | 231 (25.7) | 277 (24.5) | 287 (24.9) |
| Total | 772 (100) | 973 (100) | 1008 (100) | 898 (100) | 1129 (100) | 1152 (100) |
| Room temperature (°C), mean (SD) | 28.7 (3.4) | 30.2 (3.4) | 30.1 (2.6) | 28.3 (3.5) | 29.8 (3.2) | 29.6 (2.6) |

*Characteristics are presented by thirds (low, medium and high level) of ranked NTLI.
APCAPS, Andhra Pradesh Children and Parents Study.

intervention status. In a second sensitivity analysis, we removed three borderline outlier villages identified in analyses of BMI, SBP and FPG. The results of this sensitivity analysis were consistent with the overall and age-stratified main analyses for BMI, SBP and FPG (see online supplemental figures 4, 5 and tables 4, 5). However, results for SBP (not adjusted for room temperature) were not statistically significant at the 95% level. Online supplemental figure 6 illustrates at which levels of NTLI the fully adjusted sensitivity analysis predicted the greatest probability of differences in associations of NTLI and CVD risk factors by age group.

## DISCUSSION

Among 6236 participants, NTLI was associated with increased model predicted mean BMI, SBP and serum fasting LDL, but not FPG after controlling for confounders. We expected younger individuals to more readily adopt new lifestyles and behaviours associated with urbanisation and CVD risk factors than older individuals.[9 44] This was however not apparent from the data. The observed stronger associations of NTLI with BMI and SBP in older age groups were likely attributable to advancing age.[32] Gender did not modify associations.

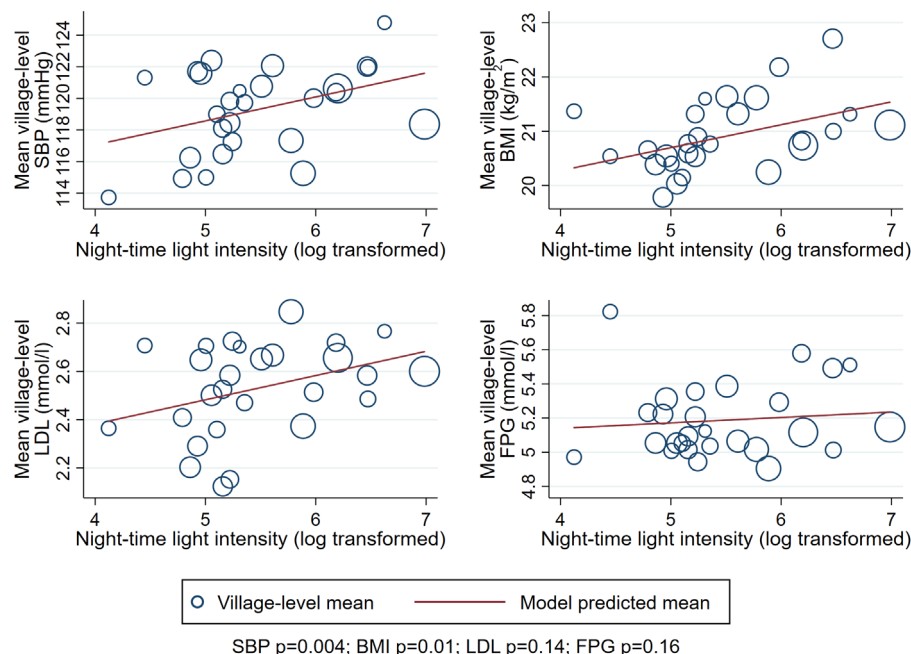

SBP p=0.004; BMI p=0.01; LDL p=0.14; FPG p=0.16

**Figure 2** Crude associations of night-time light intensity (NTLI) with CVD risk factors among APCAPS adults, 2010–2012 (n=5937). Model predicted means were derived from multilevel linear regression with clustering by household and village, using individual-level outcome data. Marker size proportional to village size. APCAPS, Andhra Pradesh Children and Parents Study; BMI, body mass index; CVD, cardiovascular disease; FPG, asting plasma glucose; LDL, low-density lipoprotein; SBP, systolic blood pressure.

## Comparison with studies using NTLI

Few studies have used NTLI data to explore relationships of urbanisation and CVD risk factors. A cross-sectional study set in rural and urban Korea (n=8526)[46] and a multi-country study of nationally representative data (n=130 counties),[47] investigated associations of NTLI with BMI. Consistent with our results on BMI, increasing level of NTLI was associated with rising levels of overweight and obesity in both studies. A cross-sectional study from India, which analysed NTLI data from NASA's Visible Infrared Imaging Radiometer Suite,[48] corroborates our adjusted results for SBP, although the included set of covariates differed between our studies. We did not find any studies on relationships between NTLI and LDL or FPG.

## Comparison with studies using multicomponent urbanisation scores

Our findings on BMI and SBP agree with cross-sectional studies from South India[15] (n=3705) and Sri Lanka (n=4485),[18] which used multicomponent scores to estimate urbanisation levels in multiprovince/state representative samples. Although we cannot compare our results directly to studies that use urbanisation scores (as opposed to NTLI), the direction of effects from low to medium (and high) urbanisation levels are consistent with our findings from an early-stage urbanisation population and support evidence of an association. We found no studies exploring relationships between urbanisation score and FPG, whereas findings from Sri Lanka[18] and China[49] suggest a greater likelihood of diabetes at high

but not medium level of urbanisation score (ranging from rural areas to large cities). It is possible that we did not observe an association of urbanisation level with FPG in our study because adverse effects of urbanisation on FPG do not manifest until a medium level of urbanisation is exceeded. However, restricting our analysis to non-diabetics may also have contributed to the null results for FPG, particularly as excluded individuals were on average older and had higher BMI.

## Comparison with rural–urban and migrant studies

Measuring urbanisation is challenging, particularly in settings where population and environment data are sparse.[22] As a result, relationships of urbanisation and CVD risk factors are often inferred from rural–urban comparisons and migrant studies. Several of these studies focus on or include Asian populations. An Indian migration study of 3537 sibling pairs suggests that mean BMI is lower among rural than migrant and urban populations. Patterns were similar for mean SBP, LDL and FPG among men but not women.[45] In a different analysis of the same study, cumulative exposure to urban environments additionally increased mean BMI of both genders as well as SBP and fasting glucose of men, while LDL remained unaffected.[13] SBP (as the only outcome) was on average higher among men than women in our sample; however, gender did not modify associations. A cross-sectional study from Southern India suggests that mean BMI, SBP and the prevalence of diabetes were lower among periurban residents of both genders than among

**Table 2** Crude and adjusted associations of NTLI with CVD risk factors among APCAPS adults, 2010–2012 (n=5937)

| | N | Model predicted crude mean (95% CI) at the lowest NTLI (61.7 (4.1 on the log scale)) | Model predicted crude mean (95% CI) at the highest NTLI (1081.1 (7.0 on the log scale)) | P value | N | Model predicted age and gender adjusted mean change (95% CI) with increasing NTLI* β (95% CI) | P value | N | Model predicted fully adjusted mean change (95% CI) with increasing NTLI*†§ β (95% CI) | P value |
|---|---|---|---|---|---|---|---|---|---|---|
| Systolic blood pressure (mm Hg)‡ | 5752 | 117.2 (115.1 to 119.4) | 121.6 (119.2 to 123.9) | 0.04 | 5747 | 1.8 (0.4 to 3.2) | 0.01 | 5514 | 1.5 (0.3 to 2.6) | 0.01 |
| Body mass index (kg/m²) | 5924 | 20.3 (19.8 to 20.8) | 21.5 (21.0 to 22.1) | 0.01 | 5920 | 0.4 (0.1 to 0.8) | 0.01 | 5682 | 0.3 (0.03 to 0.7) | 0.05 |
| Fasting serum LDL (mmol/L) | 5287 | 2.4 (2.2 to 2.6) | 2.7 (2.5 to 2.9) | 0.06 | 5285 | 0.1 (0.01 to 0.2) | 0.04 | 5051 | 0.1 (−0.003 to 0.2) | 0.06 |
| Fasting plasma glucose (mmol/L)§ | 5328 | 5.1 (5.0 to 5.3) | 5.2 (5.0 to 5.4) | 0.61 | 5326 | 0.04 (−0.1 to 0.2) | 0.52 | 5094 | −0.002 (−0.1 to 0.1) | 0.97 |

Model predicted means (95% CIs), β coefficients (95% CIs) and p-values were obtained from mutilevel linear regression models with clustering by household and village (using individual-level outcome data).
Participants were excluded if medicated for hypertension‡ or diabetes§.
*Mean change per integer increase in log transformed NTLI.
†Adjusted for age, gender, caste, religion, marital status and survey season.
β, beta-coefficient; APCAPS, Andhra Pradesh Children and Parents Study; CVD, cardiovascular disease; LDL, low-density lipoprotein; NTLI, night-time light intensity.

town and city dwellers.[50] Urban versus rural residence was also associated with higher prevalence of diabetes[49] and mean LDL in China[51] after adjusting for important confounders. Two large studies from 56[8] and 36[52] countries (n=878 000 and 148 579, respectively) across Africa, the Middle East, Asia, Americas, the Caribbean and USA suggest that overweight (BMI ≥25 kg/m²) is considerably more common among urban than rural women, particularly in lower income countries.[8] In a subsample from India (n=7608), the prevalence of overweight was almost fivefold among urban (26.4 %) compared with rural women (5.6 %).[52]

### Strengths and limitation

A major contribution of our study to the existing evidence was exploring CVD risk factors in a transitioning population in early-stage urbanisation, that is, 27 rural villages that have experienced rapid and uneven urbanisation over time and thus represented a continuum of urbanisation levels from rural settlement to medium size town at the time of survey in 2010–2012. The village-level NTLI has been validated as a standardised and comparable proxy of urbanisation level, reflecting extent of build-up area, population density, economic activity and energy consumption worldwide.[24–27] It may further be a more objective measure than the survey-based data used for multiple-component urbanisation scales, which are at risk of both interviewer and reporting biases. The NTLI data have gained popularity for exploring associations of urbanisation level and artificial light at night with human health, particularly cancer,[53–61] overweight and obesity.[46 47 62] The NTLI time series has further been suggested a valid proxy for urbanisation dynamics over time,[24 27] especially in rapidly developing countries with high urban growth rates such as China, India and Brazil.[27] Recent efforts to calibrate the NTLI time series has shown great promise for studying urbanisation dynamics in these settings[28] and offer a unique resource for health research worldwide. We calibrated the NTLI data using a robust and reliable semiautomatic method to reduce systematic bias from satellite discrepancies and enable more efficient and precise comparisons of results between studies from different regions and time points.[28] However, the semiautomatic method could introduce new bias from manual processing.[28] The overall agreement of our result with existing evidence (using different measures of urbanisation level) supports the utility of NTLI to predict urbanisation-related changes in CVD risk factors. An important limitation of our study was use of a village level exposure with individual-level outcomes. Future studies could reduce the potential for bias related to the ecological fallacy by allowing for intrasettlement variations in NTLI. The large sample size and use of hierarchical statistical methods should reduce overestimation of the magnitude of associations from risk factor clustering by sampling units (villages and households). The comparisons of excluded and included participants did however warrant some caution of generalising the results

to the general population. Trained field and clinical staff followed standardised protocols and collected information in the local language. A comprehensive list of factors were measured and explored as potential confounders and effect modifiers in our study; however, some residual confounding from unmeasured factors may remain. Data for a number of covariates were self-reported, and some recall or reporting bias cannot be rule out. We did not adjust our results for sociodemographic factors such as socioeconomic status, education and occupation because these were considered important factors on the causal pathway between urbanisation and CVD risk factors in the current population, who remained at early stages of urbanisation. Most urbanisation in India has predominantly been driven by the reclassification of rural settlements as they grow and develop.[38] Although adversity in rural areas, for example, poor opportunities for education and paid work, also contribute to urban growth through internal migration in India, the migration flow is mainly from rural to highly urbanised areas.[38 63] The 27 villages analysed in the current study remained at early stages of urbanisation and represented a continuum from rural settlement to medium-sized town. We hypothesised that participants who, for example, pursued better education or work opportunities, migrated out of the study area (to urban areas or abroad) were lost to follow-up and did not confound the results by migrating to higher urbanisation level villages within the study area. We believe that the contribution of sociodemographic factors, such as socioeconomic status, education and occupation, to the association of urbanisation on CVD risk factors in populations in early-stage urbanisation should be investigated using causal methods. Finally, the cross-sectional design of the present study prevented us from inferring causality of the results as well as from exploring the relative contribution of different urbanisation elements to CVD risk.

### Hypothesised pathways linking urbanisation and CVD risk factors

There are a number of plausible pathways through which urbanisation may deteriorate cardiovascular health. Physical and build environments change with urbanisation and in turn influence the level of exposure to noise and air pollution, green spaces and infrastructure.[22 48 64 65] Urbanisation is also associated with changes in occupation patterns and socioeconomic status[15 64]; social norms, cohesion and support; as well as personal tastes and preferences.[22 48 65] Together with increased availability, access and advertising of commodities, these transitions promote lifestyle changes that are strongly associated with CVD risk, for example, 'nutrition transition' towards diets high in fat, sugar and salt as well as processed and convenience foods; physical inactivity during work, leisure time, transport and household chores; and tobacco use and alcohol consumption.[9 18 22 64 66] Cumulative exposure to urban environments additionally appears to increase CVD risk.[13 17] Recent studies suggest that increased exposure to artificial light at night observed with increasing urbanisation level supresses melatonin production, disrupts circadian rhythm and leads to physiological and behavioural changes associated with CVD.[47] A number of studies focusing on artificial light at night (other than NTLI) without accounting for urbanisation support this mechanism.[54 67–72] At the same time, greater level of urbanisation is also associated with a number of changes that may improve CVD health. Greater level of urbanisation is associated with greater levels and higher quality of education; improved opportunities for paid work, particularly for women; higher socioeconomic status and better living conditions; and better access to health, social services and safe food, which have great potential to improve cardiovascular health.[73–78] Although these pathways have been studied extensively, no study has to our knowledge attempted to quantify their relative harmful and beneficial contributions.

### Implications

Our findings indicate that the harmful impacts of increasing urbanisation level on SBP and BMI, well-established leading modifiable risk factors for CVDs, outweigh any beneficial impacts even with moderate increases in urbanisation level. These findings suggest an opportunity for policy makers and urban planners to help curb expected increases in urbanisation-related CVDs by addressing underlying and immediate risk factors for elevated SBP and BMI during early stages of urbanisation. We will however need to understand better and quantify the implicated pathways in different settings in order to identify cost-effective prevention strategies.

### CONCLUSION

We found that the NTLI data, a novel measure of urbanisation, is a useful tool to predict changes in important urbanisation-related CVD risk factors where objective data on urbanisation factors are not readily available. The observed consistent associations of moderate increases in urbanisation level (measured by NTLI) with CVD risk factors provide an important early warning for the potential progression of the CVD burden in South India with continued urbanisation. To curb expected large public health impacts from continued and rapid urbanisation in India, further research is warranted to identify the most important mechanisms. To enable targeting of cost-effective prevention strategies, we suggest that future studies focus on transitioning populations and use causal methods to better understand potentially important pathways.

**Author affiliations**
[1]Department of Population Health, London School of Hygiene & Tropical Medicine, London, UK
[2]Department of Geography & Environment, University of Southampton, Southampton, UK
[3]Department of Medical Statistics, London School of Hygiene & Tropical Medicine, London, UK
[4]Department of Geography, The University of Sheffield, Sheffield, UK

[5]Department of Non-communicable Disease Epidemiology, London School of Hygiene & Tropical Medicine, London, UK

**Contributors** All authors contributed substantially to study conception and manuscript revision and approved the final manuscript for publication. TBS and JG cleaned and managed the APCAPS data in collaboration with the APCAPS team. RW extracted, calibrated and managed the night-time light intensity data; JG advised on statistical methods. TBS conducted the literature search, data analysis and interpretation and drafted the manuscript; SK is the director of the APCAPS.

**Funding** This work was supported by Wellcome Trust (www.wellcome.ac.uk/) Grant number 083707 and the Bloomsbury PhD Studentship (http://www.bloomsbury.ac.uk/) through LSHTM, London, UK. ADD and BS acknowledge the Sustainable and Healthy Food Systems (SHEFS) programme supported by the Wellcome Trust's 'Our Planet, Our Health' programme (grant no. 205200/Z/16/Z).

**Competing interests** None declared.

**Patient consent for publication** Not required.

**Ethics approval** Ethics approvals were obtained from Public Health Foundation of India, New Delhi, India; National Institute of Nutrition, Hyderabad, India; and the LSHTM, London, UK. Approvals were obtained from all village heads and their committees. Study participants provided written informed consent, or a witnessed thumbprint if illiterate, prior to study start.

**Provenance and peer review** Not commissioned; externally peer reviewed.

**Data availability statement** The dataset analysed in the current study is available from the APCAPS (http://apcaps.lshtm.ac.uk/) on reasonable request.

**ORCID iD**

Tina Bonde Sorensen http://orcid.org/0000-0003-1517-6159

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
