## [Reviewer comments · BMJ Open]

ARTICLE DETAILS

TITLE (PROVISIONAL)	IS NIGHT-TIME LIGHT INTENSITY ASSOCIATED WITH CARDIOVASCULAR DISEASE RISK FACTORS AMONG ADULTS IN EARLY STAGE URBANISATION IN SOUTH INDIA? A CROSS SECTIONAL STUDY OF THE ANDHRA PRADESH CHILDREN AND PARENTS STUDY
AUTHORS	Sorensen, Tina Bonde; Wilson, Robin; Gregson, John; Shankar, Bhavani; Dangour, Alan; Kinra, Sanjay

VERSION 1 – REVIEW

REVIEWER	Qiqiang He School of Health Sciences, Wuhan University
REVIEW RETURNED	27-Dec-2019

GENERAL COMMENTS	1. Why did the study use only the third wave if the Andhra Pradesh Children and Parents Study (APCAPS)? It is stated that "a third APCAPS survey wave included all available HNT index children, their parents, and siblings (n=6944)". Is that only the third wave included data of parents? Please clarify this. Also, the sample size is confusing. Is the number 6944 a sum of children, their parents, and siblings, which seems problematic? 2. Please briefly introduce the sampling strategy or at least cite the method of the trial. Though it is stated to be "a large cluster-randomized population-based sample", it is also stated to use data of parents of new-born, which is confusing. Is this village-level representative data or did all new-born babies during the time were included? 3. Why did the study choose certain biomarkers from a pool of CVD risk factors detected excluding triglyceride? And please consider use one comprehensive variable of cardiometabolic risk, like metabolic syndrome or a continuous risk factor derived from factor analysis. 4. The night-time light intensity is a country-level variable (second level), while socio-economic status (like education and occupation) are individual-level variables (first level). Therefore, I think it is proper to adjust them simultaneously in a hierarchical model.
---

REVIEWER	Na-Jin Park University of Pittsburgh, USA
REVIEW RETURNED	23-Mar-2020

GENERAL COMMENTS	BMJ Open
----------

	Night-time light intensity consistently associated with cardiovascular disease risk factors: a cross sectional study of the Andhra Pradesh children and parents' study, South India Review: Major revision  - The whole paper needs paragraphing beyond headings and subheadings. For example, "Background" is consisted of 1 paragraph, which can be divided to 3-4 paragraphs with themes as follows: Cardiovascular disease (CVD) risk and risk factors; Urbanization and night-time light intensity (NTLI); night-time light and CVD risk; and the objective of the study to fill up the gaps in the literature). - The Methods section needs revision for better clarity about the APCAPS cohort and the use of third follow-up data in 2010-2012, not 1992 or 2003. I would like to see a brief, overall description of data collections on study variables including NTLI in a separate section of "procedure" or you can incorporate into appropriate sections of the measures. - Under the Measures, I recommend organizing it by outcomes, main predictor(s), and covariates. Please focus on BMI, SBP, LDL and FPG in explaining biochemical assays. Current information is confusion. Room temperature seems something to control, which needs to be included as a covariate with an appropriate justification. - Please address how NTLI levels were quantified with units and scores (values) and definitions of urbanization and/or NTLI (e.g., mild or early stage urbanization, transition, etc.) for the study. What other measures were used in previous studies, including pros and cons compared to the current NTLI? It would be a good topic for discussion. - Some information on measures in statistical analysis should be moved to the section of measures (instead of "data"). - Avoid using unnecessary abbreviations such as DMSP (p14 line 272). Consider to use the full term for NTLI. - The separate section of "Possible Mechanisms" seems bits of stretch from this study focusing on an individual effect of each risk factor, but not on any biobehavioral mechanisms. It would be better to incorporate into "background" or "discussion". - Although the original APCAPS cohort-related information is important for the study because it is the source of the cohort, this secondary-analysis serves different purposes from the original to explore the relationship between urbanization and CVD in adults in a wide range of age from the South India. Authors have to present this paper in the line of the current cross-sectional study, not on the original cohort study. For example, the title including the cohort name seems not necessary to display the major message of the study. Instead, I would recommend: night-time light intensity associated with CVD risk factors in adults in early stage urbanization in South India. "Consistently" indicates multiple assessments and more evidence available, which is not true. - In comparing with other studies, consider to look at early stage urbanization studies consistent with your study. Or contrast with some studies without detailed level of urbanization. That would be more interesting and relevant synthesis.
--	--

REVIEWER	Alipasha Meysamie, MD, MPH Tehran University of Medical Sciences IR Iran
REVIEW RETURNED	14-Jun-2020

GENERAL COMMENTS	One important limitation of the study seems it should be considered is referring to ecological context of NTLI because it has been considered for all persons who are living in one village and have not been measured individually, so ecological fallacy may play a role in this study. In table one in the last row "Room temperature (°C), mean (95% CI)" seems to be mean (SD) and not 95% CI Supplementary Fig. 4 Sensitivity analysis: in 2nd figure SBP has been missed. Some confidence intervals need to be rechecked e.g. Tab 2 last row Fasting plasma glucose (mmol/l) 5.328 "5.1 (5.0, 5.3)". Accept with minor revision
---

VERSION 1 – AUTHOR RESPONSE

Reviewer(s)' Comments to Author:

Reviewer: 1

Reviewer Name: Qiqiang He

Institution and Country: School of Health Sciences, Wuhan University

Please state any competing interests or state 'None declared': None declared

1. Why did the study use only the third wave if the Andhra Pradesh Children and Parents Study (APCAPS)?

It is stated that "a third APCAPS survey wave included all available HNT index children, their parents, and siblings (n=6944)". Is that only the third wave included data of parents? Please clarify this.

Also, the sample size is confusing. Is the number 6944 a sum of children, their parents, and siblings, which seems problematic?

Response: We apologise for a lack of clarity on these important details. The section on 'Study design, setting and participants' has now been revised to improve clarity on the nature of the APCAPS cohort over time and to explain our rationale for using the third survey wave. The flow chart in Fig. 1 has also been revised and a footnote added to the 'Study design, setting and participants' section to more clearly specify the number of index children, parents and siblings included in the APCAPS.

2. Please briefly introduce the sampling strategy or at least cite the method of the trial.

Though it is stated to be "a large cluster-randomized population-based sample", it is also stated to use data of parents of new-born, which is confusing. Is this village-level representative data or did all new-born babies during the time were included?

Response: We have now referenced the publications that provide the details on the HNT on which the APCAPS was based as well as the establishment and expansion of the APCAPS.

3. Why did the study choose certain biomarkers from a pool of CVD risk factors detected excluding triglyceride? And please consider use one comprehensive variable of cardiometabolic risk, like metabolic syndrome or a continuous risk factor derived from factor analysis.

Response: We selected systolic blood pressure, body mass index, fasting low-density lipoprotein and fasting plasma glucose because of their well-established and strong associations with CVDs, their modifiable nature, and because they are widely reported in the urbanisation literature. The advantage of analysing CVD risk factors that are widely reported in existing urbanisation literature is that comparing our results to published evidence offered insights to the utility of using a globally available,

standardised and free novel proxy of continuous urbanisation levels (NTLI) to study cardiovascular health. This opportunity would have been lost if we also used a novel study outcome, such as the suggested comprehensive composite variable derived from factor analysis, or cardiometabolic disease, which to the best of our knowledge, is not yet widely investigated in the urbanisation literature. We also chose to focus on continuous outcomes because they enabled the exploration of linear relationships between increasing level of NTLI and CVD risk factors across a continuum of urbanisation levels that was unique to this study.

4. The night-time light intensity is a country-level variable (second level), while socio-economic status (like education and occupation) are individual-level variables (first level). Therefore, I think it is proper to adjust them simultaneously in a hierarchical model.

Response: We recognize the potential confounding effect of socio-economic status, occupation and education in analyses of urbanisation level on CVD risk factors in populations at high levels of urbanisation. However, in the current population that remained in early stage urbanisation, we consider it more likely that these variables are on the causal pathway from exposure to outcome and therefore do not think the proposed hierarchical model would be appropriate. We have added a paragraph in the discussion to expand on the justification for this choice.

Reviewer: 2

Reviewer Name: Na-Jin Park

Institution and Country: University of Pittsburgh, USA

1. The whole paper needs paragraphing beyond headings and subheadings. For example, "Background" is consisted of 1 paragraph, which can be divided to 3-4 paragraphs with themes as follows: Cardiovascular disease (CVD) risk and risk factors; Urbanization and night-time light intensity (NTLI); night-time light and CVD risk; and the objective of the study to fill up the gaps in the literature).

Response: Thank you for this suggestion which has improved the structure of our paper.

2. The Methods section needs revision for better clarity about the APCAPS cohort and the use of third follow-up data in 2010-2012, not 1992 or 2003. I would like to see a brief, overall description of data collections on study variables including NTLI in a separate section of "procedure" or you can incorporate into appropriate sections of the measures.

Response: We apologise for the lack of clarity that was also noted by Reviewer 1. The methods section has been revised as requested to more clearly describe the APCAPS cohort and the rationale for using the third survey wave (also see response to Reviewer 1's comment 1).

The revised 'Methods' section now provides more information on data collection and procedures, particularly in relation to NTLI.

3. Under the Measures, I recommend organizing it by outcomes, main predictor(s), and covariates. Please focus on BMI, SBP, LDL and FPG in explaining biochemical assays. Current information is confusion. Room temperature seems something to control, which needs to be included as a covariate with an appropriate justification.

Response: The 'Data' section has been reorganised as requested. We did not adjust the main analysis of SBP for room temperature because 45% of the sample were missing data on room temperature. Instead, we presented room temperature-adjusted analyses in the Supplementary Material. We have now clarified the extent of missing data on room temperature in the 'Methods' section and refer to the Supplementary Material for results.

4. Please address how NTLI levels were quantified with units and scores (values) and definitions of urbanization and/or NTLI (e.g., mild or early stage urbanization, transition, etc.) for the study. What other measures were used in previous studies, including pros and cons compared to the current NTLI? It would be a good topic for discussion.

Response: A subsection on defining urbanisation levels and early stage urbanisation has been added to the 'Methods' section. In the 'Data' section, we now also expand on the collection of the NTLI data, its unitless nature, and the derivation of NTLI-based urbanisation levels. We hope that the restructuring and paragraphing of the 'introduction' and 'Discussion' sections satisfactorily bring out the discussions on urbanisation measures used in existing literature and their advantages and disadvantages compared with the NTLI measure.

5. Some information on measures in statistical analysis should be moved to the section of measures (instead of "data").

Response: Changed as requested.

6. Avoid using unnecessary abbreviations such as DMSP (p14 line 272). Consider to use the full term for NTLI.

Response: Unnecessary abbreviations have been removed. To avoid considerably lengthening the manuscript we have kept the abbreviation for night-time light intensity (NTLI).

7. The separate section of "Possible Mechanisms" seems bits of stretch from this study focusing on an individual effect of each risk factor, but not on any biobehavioral mechanisms. It would be better to incorporate into "background" or "discussion".

Response: We have amended the subheading for the section on 'Possible mechanisms' to 'Hypothesised pathways linking urbanisation and CVD risk factors' to be clear that we did not study the mechanisms of the association of urbanisation level on CVD risk factors.

8. Although the original APCAPS cohort-related information is important for the study because it is the source of the cohort, this secondary-analysis serves different purposes from the original to explore the relationship between urbanization and CVD in adults in a wide range of age from the South India. Authors have to present this paper in the line of the current cross-sectional study, not on the original cohort study. For example, the title including the cohort name seems not necessary to display the major message of the study. Instead, I would recommend: night-time light intensity associated with CVD risk factors in adults in early stage urbanization in South India. "Consistently" indicates multiple assessments and more evidence available, which is not true.

Response: The section on 'Study design, setting and participants' has been revised to better align to the third survey wave and the composition of its sample. We also thank you for the helpful title suggestion which has improved the title of our study.

9. In comparing with other studies, consider to look at early stage urbanization studies consistent with your study. Or contrast with some studies without detailed level of urbanization. That would be more interesting and relevant synthesis.

Response: Most evidence on associations of urbanisation with CVD risk factors comes from rural-urban comparisons that include city populations. To our knowledge, only a limited number of studies have used other measures of urbanisation (e.g. remote sensing data or urbanisation scales) in studies of the CVD risk factors reported in our study. In addition, none of these studies are (to our

knowledge) from populations in early stage urbanisation. We have added additional information on the range of urbanisation levels analysed in existing literature, and sought to strengthen the discussion of our results in the context of existing literature by comparing our results more explicitly to results observed at low to medium levels of urbanisation in existing literature.

Reviewer: 3

Reviewer Name: Alipasha Meysamie, MD, MPH

Institution and Country:

Tehran University of Medical Sciences

IR Iran

1. One important limitation of the study seems it should be considered is referring to ecological context of NTLI because it has been considered for all persons who are living in one village and have not been measured individually, so ecological fallacy may play a role in this study.

Response: We acknowledge this shortcoming and have added a statement on this limitation to the discussion.

2. In table one in the last row "Room temperature (°C), mean (95% CI)" seems to be mean (SD) and not 95% CI

Response: Corrected as requested.

3. Supplementary Fig. 4 Sensitivity analysis: in 2nd figure SBP has been missed. Some confidence intervals need to be rechecked e.g. Tab 2 last row Fasting plasma glucose (mmol/l) 5.328 "5.1 (5.0, 5.3)".

Response: We have corrected Supplementary Fig. 4 as requested and rechecked that confidence intervals in Table 2 are correct.

VERSION 2 – REVIEW

REVIEWER	Alipasha Meysamie Tehran university of medical sciences Iran
REVIEW RETURNED	10-Aug-2020
GENERAL COMMENTS	Authors considered comments of the previous round of review in an acceptable form in the revision. Decision: Accept